# MOLAR: MULTIMODAL LLMS WITH COLLABORATIVE FILTERING ALIGNMENT FOR ENHANCED SEQUENTIAL RECOMMENDATION

## ABSTRACT

Large language models (LLMs) have significantly advanced sequential recommender (SR) systems by their strong contextual understanding and long-range dependency modeling capabilities. However, we hold that LLMs alone often yield suboptimal recommendations due to their inherent lack of explicit collaborative filtering modeling, relying predominantly on textual content while neglecting item ID-based interaction information. To address these limitations, we propose **Molar**, a **M**ultim**o**dal large **la**nguage sequential **r**ecommendation framework that integrates multiple content modalities with ID information to capture collaborative signals effectively. Molar employs an MLLM to generate unified item representations from both textual and non-textual data, facilitating comprehensive multimodal modeling and enriching item embeddings. Additionally, it incorporates collaborative filtering signals through a post-alignment mechanism, which aligns user representations from content-based and ID-based models, ensuring precise personalization and robust performance. By seamlessly combining multimodal content with collaborative filtering insights, Molar captures both user interests and contextual semantics, leading to superior recommendation accuracy. Extensive experiments validate that Molar outperforms traditional and LLM-based baselines, highlighting its strength in utilizing multimodal data and collaborative signals for sequential recommendation. The source code is available here [1].

## 1 INTRODUCTION

In the context of information overload, recommender systems (Resnick & Varian, 1997; Koren, 2008) have become one of the most effective information filtering mechanisms across various online applications, such as e-commerce, advertising, and online video platforms. Among them, sequential recommender (SR) systems (Wang et al., 2019; Zhou et al., 2018; Kang & McAuley, 2018) have gained increasing attention due to their significant advantages in capturing dynamic user interests compared to traditional collaborative filtering methods. Currently, mainstream approaches predominantly adopt ID-based deep learning strategies (Koren et al., 2009; Goldberg et al., 1992), ranging from matrix factorization to deep sequence neural networks, where users and items are converted into unique identifiers and encoded through embedding tables. By training on historical interaction data, these methods can capture the sequential behavioral patterns of users.

Though achieving success, these methods still fall short in recommendation performance due to the sparsity of item interactions. ID-based strategies perform poorly in cold-start scenarios, and shallow neural networks limit their ability to effectively model users' complex and dynamic interests. Fortunately, various content features such as visual appearance and textual descriptions related to each item can be utilized to mitigate these shortcomings in sequential recommendation. To integrate content features and enhance SRS performance, researchers have proposed several attempts in prior studies (Hidasi, 2015; Zhang et al., 2019). However, we argue that the performance improvements of these methods in mining content features for sequential recommendation are limited for two main reasons: (1) Insufficient integration of multiple content features, such as misalignment between image and text features, leading to information loss during content integration training; (2) Neglecting

---

[1] https://anonymous.4open.science/r/Molar-8B06/

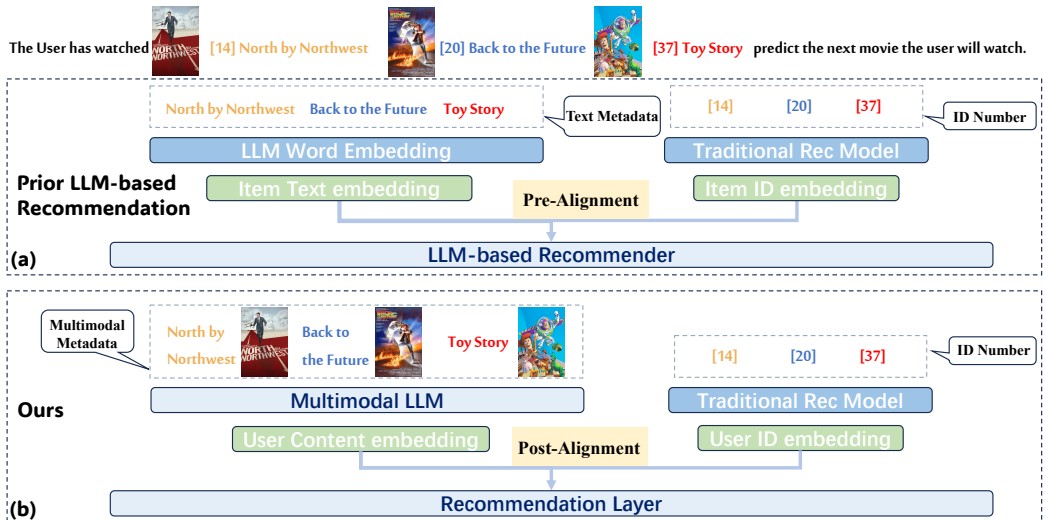

Figure 1: **Comparison of LLM-based recommendation methods and our Molar**. (a) Existing methods prematurely integrate ID and text modalities into the LLM, leading to limited utilization of multimodal content features. (b) Our approach first processes text and non-text modalities through the LLM to generate rich multimodal representations and then incorporates ID information via post-alignment, ensuring a better balance between multimodal content and collaborative signals.

the semantic differences between ID-based user embeddings and content-based user embeddings. ID-based embeddings can capture implicit collaborative signals among users, while content-based embeddings typically encode highly specific item descriptions, reflecting user preferences from different perspectives. Ignoring this complementary information results in suboptimal performance.

Recent advances in large language models (LLMs) (Zhao et al., 2023; Luo et al., 2024) have opened new possibilities for sequential recommendation. With their powerful sequence modeling and multimodal understanding capabilities, LLMs have been explored in two main directions. The first approach (Zhang et al., 2023b; Ren et al., 2024; Friedman et al., 2023) reframes SR as a natural language processing task, allowing LLMs to interpret user sequences and generate recommendations based on their language understanding. The second approach (Ning et al., 2024; Liao et al., 2023) combines LLMs with traditional SR models by integrating ID and text modalities into the LLM backbone at an early stage, as shown in Figure 1a. However, these methods face critical limitations: (1) inadequate integration of multimodal features, leading to information loss from non-textual modalities or misalignment between textual and visual features; and (2) suboptimal utilization of traditional SR models, where early fusion of ID information can cause LLMs to learn shortcuts, overshadowing collaborative filtering signals. These challenges result in a failure to fully exploit the multimodal features of items and the potential of collaborative filtering in SR.

To address these issues, this paper introduces the **Molar**, a **M**ultim**o**dal large **la**nguage sequential **r**ecommendation framework. We make three main contributions in this paper. *First*, we propose a Multimodal Item Representation Model (MIRM) based on a multimodal large language model (MLLM) to extract item features from both textual and non-textual modalities. By fine-tuning MIRM on multimodal data, we ensure robust and consistent item embeddings. *Second*, we design a Dynamic User Embedding Generator (DUEG) that leverages these item embeddings to model user interests and predict future behaviors. This allows for effective user modeling in complex multimodal scenarios. *Third*, we introduce a post-alignment contrastive learning mechanism that aligns collaborative signals from ID-based and content-based models, preserving the strengths of both representations. This mechanism ensures semantic alignment between user embeddings while enhancing accuracy.

Experiments conducted on multiple datasets demonstrate the effectiveness of Molar. The framework significantly outperforms traditional SR models and state-of-the-art LLM-based methods, achieving superior results in recommendation accuracy and robustness. Our results show that Molar captures user interests more comprehensively by combining multimodal content and collaborative filtering signals, leading to consistent performance improvements across diverse scenarios.

Our contributions are summarized as follows:

- We propose a Multimodal Item Representation Model (MIRM) to extract robust item embeddings by leveraging multimodal content, including both textual and non-textual modalities, ensuring comprehensive item feature modeling.

- We design a Dynamic User Embedding Generator (DUEG) to model user preferences using multimodal item embeddings, enabling dynamic and accurate user interest prediction.

- We introduce a post-alignment contrastive learning mechanism to integrate collaborative filtering signals from ID-based and content-based models, preserving their complementary strengths and enhancing performance.

## 2 RELATED WORK

### 2.1 LLM-BASED RECOMMENDATION.

The success of LLMs such as GPT4 OpenAI et al. (2024) and LLaMA Grattafiori et al. (2024) has sparked extensive exploration of their application in recommendation systems. Firstly, LLMs are used to enhance user or item information, such as aligning semantic spaces for user and item profiling or generating training signals for cold-start items Xi et al. (2024); Ren et al. (2024); Zhang et al. (2024b). Secondly, some studies convert recommendation data into a conversational format, leveraging LLMs' instruction-following capabilities to predict user behavior Friedman et al. (2023); Bao et al. (2023); Zhang et al. (2023a). Lastly, LLMs are adapted for recommendation tasks by combining ID-based item embeddings with textual features for hybrid prompting or using them for multi-class classification and regression for rating prediction Ning et al. (2024); Liao et al. (2023). Although these methods demonstrate the potential of LLMs, improvements over traditional recommendation models remain limited. Prior methods either overlook traditional ID-based models by focusing only on text or introduce ID modalities too early, reducing the effectiveness of collaborative filtering signals during LLM training. Unlike these approaches, we propose a post-alignment mechanism to fuse ID-based collaborative information later in the process, preserving LLMs' world knowledge while retaining essential collaborative information.

### 2.2 MULTIMODAL LARGE LANGUAGE MODELS.

Recent advancements Peng et al. (2023); Zhang et al. (2024c); Yin et al. (2024) in multimodal pre-training have significantly improved task performance but at the cost of increased computational demands. To address this, researchers are leveraging pre-trained unimodal models, particularly large language models (LLMs) Kasneci et al. (2023), to develop Multimodal Large Language Models (MLLMs) that integrate language with other modalities. The primary challenge lies in achieving effective model collaboration, with a focus on aligning modalities and understanding human intent. MLLMs like GPT-4 OpenAI et al. (2024) and Gemini Team et al. (2023) have demonstrated exceptional capabilities in multimodal comprehension. Some studies Wang et al. (2024); Lu et al. (2024); Zhang et al. (2024a) have concentrated on integrating LLMs with visual encoders. Inspired by these prior studies, we have developed Molar, which utilized MLLMs to align and fuse multimodal information to enhance sequential recommendation.

### 2.3 MULTIMODAL SEQUENTIAL RECOMMENDATION

Sequential Recommenders (SRs) have advanced from traditional matrix-based models to complex neural network architectures. Early approaches, such as Factorizing Personalized Markov Chains (FPMC), combined matrix factorization and Markov chains to capture sequential behavior Rendle et al. (2010). The shift toward neural models began with GRU4Rec, which used gated recurrent units for session-based recommendations Tan et al. (2016). Later, SASRec introduced self-attention mechanisms to address long-term dependencies in user-item interactions Kang & McAuley (2018), while BERT4Rec enhanced SRs with transformers, offering deep bidirectional training for improved performance Sun et al. (2019). Additionally, multimodal SRs have emerged, incorporating extra contextual information to enhance recommendation accuracy. Fusion strategies in SRs are typically divided into early, late, and hybrid approaches Hu et al. (2023). Early fusion includes invasive methods that merge multiple modalities at the input level, such as concatenation and gating Lei et al. (2019); Tang & Wang (2018). Non-invasive early fusion uses attention mechanisms to combine

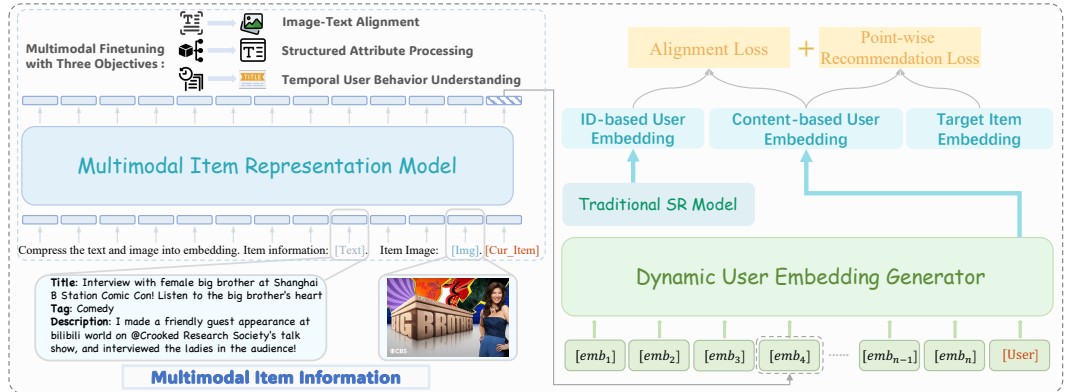

Figure 2: **Illustration of the Molar framework.** The multimodal item representation model processes multimodal item information to generate item embeddings, while the dynamic user embedding generator models user embeddings based on interaction histories for next-item prediction.

attributes before processing Liu et al. (2021). In contrast, late fusion combines feature sequences from different modules at the final stage Zhang et al. (2019); Ji et al. (2020); Du et al. (2023). Hybrid fusion methods offer flexibility by combining modality fusion and sequential modeling through inter-modality relationship evaluation Hu et al. (2023).

## 3 METHODS

### 3.1 PROBLEM FORMULATION

We tackle the task of sequential recommendation, where the goal is to predict the next item $I_{n+1}$ that a user $u \in U$ is likely to interact with, given their historical interaction sequence $H_u = \{I_1, I_2, \ldots, I_n\}$ arranged in chronological order. Each item $I_i \in I$ comes with multimodal information, such as titles, textual descriptions, and images. Our approach leverages this multimodal information to improve the prediction accuracy of the next interaction.

### 3.2 MOLAR OVERVIEW

Traditional recommendation systems based on LLMs often suffer from inefficiencies when handling extensive user histories, as transforming these histories into lengthy token sequences results in high computational costs and slower inference speeds. To address these challenges, we propose **Molar**, a decoupled framework that separates the modeling of items and users for more efficient processing. This separation allows tailored-modeling strategies for each component, improving computational efficiency and recommendation accuracy. Our framework is composed of two key modules: the multimodal item representation model (MIRM) and the Dynamic User Embedding Generator (DUEG). MIRM is designed to compress multimodal features into compact embeddings, mitigating the computational burden of lengthy token sequences. DUEG then utilizes these embeddings to build user representations that capture dynamic user preferences. Together, these modules enable effective multimodal feature modeling and user preference prediction, setting the foundation for robust sequential recommendation.

### 3.3 MULTIMODAL ITEM REPRESENTATION MODEL

To effectively extract and encode item features, we introduce the MIRM, denoted as $f_I$. This encoder leverages an MLLM to combine textual descriptions and images into a unified embedding representation. Although MLLMs excel in understanding text and images, their direct application to feature extraction remains limited. To address this, we append a special identifier, [Cur_Item], to the end of each item's description, guiding the model to focus on extracting relevant features.

The process begins by merging an item's textual and image attributes into a unified description $L$, augmented with a prompt to enhance model comprehension. $L$ is tokenized and processed by the MLLM, with [Cur_Item] appended at the end of the token sequence $\{t_1, t_2, \ldots, t_m, \text{[Cur\_Item]}\}$. The

model's hidden state corresponding to `[Cur_Item]` is extracted as the item's embedding:

$$\langle emb_i \rangle = f_I(txt_i, img_i), \tag{1}$$

where $\langle emb_i \rangle$ is embedding of item $I_i$, $txt_i$ is textual description, $img_i$ is associated image. To enhance the quality of representations, MIRM undergoes multimodal fine-tuning with three objectives:

**Image-Text Alignment.** Aligns visual features with textual descriptions, using item images to generate detailed descriptions. This alignment ensures that the model captures meaningful relationships between visual content and textual context, improving its ability to interpret multimodal data cohesively. The fine-tuning data for this objective (**Image-Text, IT**) uses item images as input and produces detailed textual descriptions as output.

**Structured Attribute Processing.** Converts structured metadata (e.g., price, material) into natural language descriptions for comprehensive feature encoding. This process allows the model to integrate diverse item attributes into a unified representation, enhancing its flexibility to handle heterogeneous data types. The fine-tuning data (**Structured Attributes, SA**) uses item titles and metadata (e.g., price, material, size, color) as input to generate detailed textual descriptions as output.

**Temporal User Behavior Understanding.** Captures temporal dynamics by predicting future items from historical interactions using multimodal inputs. This objective helps the model learn sequential patterns in user behavior, enabling it to better adapt to dynamic user preferences over time. The fine-tuning data (**User Behavior, UB**) consists of historical item interactions (descriptions and images) as input, with predicted next items as output.

These objectives enable MIRM to produce robust multimodal embeddings that integrate seamlessly into the subsequent user modeling process, bridging item representation with user preference.

## 3.4 DYNAMIC USER EMBEDDING GENERATOR

Building on the item embeddings generated by MIRM, we design the DUEG, denoted as $f_U$. This module constructs dynamic user representations based on their historical interactions, effectively capturing evolving preferences. Unlike MIRM, DUEG simplifies the structure by removing the word embedding layer from the MLLM, retaining the pretrained parameters for efficient embedding processing.

Given a user's historical interaction sequence $H_u = \{I_1, I_2, \ldots, I_n\}$, MIRM transforms each item $I_i$ into an embedding $emb_i$. These embeddings are then processed by DUEG, which incorporates a special `[User]` token to represent the user's dynamic preferences. This approach enables DUEG to predict the next likely item $I_{n+1}$ based on past interactions, formalized as:

$$E_u = f_U(emb_1, emb_2, \ldots, emb_n), \tag{2}$$

where $E_u$ is the user embedding. To optimize MIRM and DUEG, we introduce two loss functions:

**Point-wise Recommendation Loss.** To enhance the model's accuracy in predicting the next item, we employ a binary cross-entropy (BCE) loss function. In our training process, each positive sample is paired with a negative sample using a 1:1 negative sampling strategy. The target item embedding calculated from MIRM consists of a positive item (pos) and a negative item (neg). For each pair, we define label vector $y = [1, 0]$ and generate predicted logits $x = [x_{\text{pos}}, x_{\text{neg}}]$. The BCE loss is:

$$\mathcal{L}_{\text{bce}} = -(y \cdot \log(x) + (1 - y) \cdot \log(1 - x)) \tag{3}$$

Minimizing this loss encourages the model to assign higher probabilities to positive samples and lower probabilities to negative samples, thereby improving its ability to distinguish relevant items for accurate next-item predictions.

**Alignment Loss.** To bridge content-based and ID-based representations, we introduce a post-alignment mechanism that uses a contrastive learning objective for both embeddings after DUEG processes multimodal content. This prevents premature integration of collaborative filtering into the LLM, ensuring essential collaborative information is preserved. The contrastive learning objective is:

$$\mathcal{L}_{align} = -\frac{1}{|U|} \sum_{u=1}^{|U|} \left( \log \frac{\exp(s(E_u^{id}, E_u^{con})/\tau)}{\sum_{j \in K} \exp(s(E_u^{id}, E_j^{con})/\tau)} + \log \frac{\exp(s(E_u^{con}, E_u^{id})/\tau)}{\sum_{j \in K} \exp(s(E_u^{con}, E_j^{id})/\tau)} \right), \quad (4)$$

where $\tau$ is the temperature parameter, $K$ is the set of comparative instances containing one positive and $K$-1 negative examples. $s(\cdot, \cdot)$ denotes the cosine similarity function, and $E^{id}$ and $E^{con}$ are the ID-based user embeddings from a traditional sequential recommendation model and the content-based embeddings from DUEG, respectively.

The final training objective combines two losses, where $\alpha$ balances their contributions.:

$$\mathcal{L}_{total} = \mathcal{L}_{bce} + \alpha \cdot \mathcal{L}_{align}, \quad (5)$$

By integrating multimodal item features with collaborative signals, DUEG enhances accuracy and robustness in user preference modeling, enabling a seamless transition to recommendation generation.

## 4 EXPERIMENTS

In this section, we evaluate our proposed framework, Molar, on three real-world datasets and compare it against several baselines, including traditional sequential recommender models and state-of-the-art LLM-based methods. To assess the effectiveness of Molar, we conduct a comprehensive analysis addressing four research questions. Additionally, we investigate the impact of different DUEGs and the various input data modalities on the results. Furthermore, we perform ablation studies to investigate the impact of fine-tuning strategies and post-alignment, as well as evaluate the performance differences between our LLM-based user modeling approach and alternative methods.

### 4.1 EXPERIMENTAL SETUP

**Datasets and Evaluation Metrics.** We evaluate our approach using three standard datasets: (1) Amazon (Veit et al., 2015) , containing e-commerce interactions for clothing items; (2) PixelRec (Cheng et al., 2023), which provides multimodal image and text data for recommendations; and (3) MovieLens (Harper & Konstan, 2015), a widely used movie rating dataset. For all three datasets, we arrange the interaction sequences in sequential order. We utilize a leave-one-out approach to split the data into training, validation, and testing sets. Detailed statistics of the datasets are provided in Table 5. The evaluation metrics are Normalized Discounted Cumulative Gain (NDCG@K), Recall (Recall@K), which are evaluated on the full amount of data. The abbreviations N, and R are respectively used to denote NDCG, and Recall.

**Baselines.** FPMC (Rendle et al., 2010), FMLP-Rec (Zhou et al., 2022), GRU4Rec (Tan et al., 2016), and SASRec (Kang & McAuley, 2018) are traditional sequential recommendation models based on Markov Chains, RNN, and attention mechanisms, respectively. DuoRec (Qiu et al., 2022) employs contrastive learning to extract discriminative information for sequential recommendation. SASRec-Content is a variant of SASRec that directly utilizes content feature representations as sequence inputs. It includes three versions: text-only, image-only, and a combination of text and image. LLamaRec (Yue et al., 2023), MLLM-MSR (Ye et al., 2024), CoLLM (Zhang et al., 2023b) and HLLM (Chen et al., 2024) are sequential recommendation models based on LLMs, achieving state-of-the-art performance.

**Implementation Details.** We employ Qwen2vl-2b as the backbone model for both MIRM and DUEG. For each dataset, we create three types of data mixtures, each consisting of 10,000 data points, to fine-tune the MIRM. Additionally, we employ SASRec as the ID-based recommendation model for contrastive learning, with an embedding dimension same as the MIRM.

For all methods involving LLMs, each experiment is trained for a maximum of 5 epochs with a batch size of 128. A learning rate warm-up strategy is employed, initializing the learning rate at 1/100 of its maximum value 1e-4, and dynamically adjusting it over training steps using a cosine scheduler.

Table 1: **Performance comparison of Molar with baseline models.** The underlined values indicate the best and second-best results across all models. The abbreviations N and R represent Normalized Discounted Cumulative Gain (NDCG) and Recall, respectively. Statistically significant improvements are marked with * ($p$-value $<< 0.05$). Overall, Molar consistently achieves superior performance across datasets, showing its effectiveness in leveraging multimodal and collaborative filtering features.

| Methods | | Amazon* | | | | PixelRec* | | | | Movielens* | | | |
|---|---|---|---|---|---|---|---|---|---|---|---|---|---|
| | | N@10 | N@20 | R@10 | R@20 | N@10 | N@20 | R@10 | R@20 | N@10 | N@20 | R@10 | R@20 |
| Traditional | FPMC | 0.1037 | 0.1059 | 0.1152 | 0.1232 | 0.0107 | 0.0129 | 0.0191 | 0.0290 | 0.0907 | 0.1129 | 0.1708 | 0.2756 |
| | FMLP-Rec | 0.1089 | 0.1112 | 0.1209 | 0.1293 | 0.0112 | 0.0136 | 0.0201 | 0.0304 | 0.0953 | 0.1186 | 0.1793 | 0.2894 |
| | GRU4Rec | 0.1029 | 0.1054 | 0.1107 | 0.1190 | 0.0109 | 0.0127 | 0.0189 | 0.0284 | 0.0828 | 0.1081 | 0.1657 | 0.2664 |
| | SASRec | 0.1080 | 0.1105 | 0.1188 | 0.1281 | 0.0131 | 0.0149 | 0.0207 | 0.0311 | 0.1116 | 0.1395 | 0.2137 | 0.3245 |
| | DuoRec | 0.1281 | 0.1342 | 0.1406 | 0.1616 | 0.0147 | 0.0181 | 0.0241 | 0.0362 | 0.1530 | 0.1790 | 0.2704 | 0.3738 |
| Content-based | SASRec$_{Bert}$ | 0.1116 | 0.1130 | 0.1275 | 0.1365 | 0.0131 | 0.0161 | 0.0238 | 0.0357 | 0.1172 | 0.1465 | 0.2244 | 0.3407 |
| | SASRec$_{Vit}$ | 0.1142 | 0.1187 | 0.1237 | 0.1373 | 0.0126 | 0.0155 | 0.0211 | 0.0317 | 0.1204 | 0.1499 | 0.2295 | 0.3481 |
| | SASRec$_{Bert+Vit}$ | 0.1164 | 0.1179 | 0.1308 | 0.1437 | 0.0136 | 0.0167 | 0.0210 | 0.0315 | 0.1258 | 0.1567 | 0.2382 | 0.3599 |
| LLM-based | LLamaRec | 0.1242 | 0.1284 | 0.1378 | 0.1493 | 0.0151 | 0.0174 | 0.0239 | 0.0364 | 0.1387 | 0.1631 | 0.2478 | 0.3799 |
| | MLLM-MSR | 0.1304 | 0.1348 | 0.1447 | 0.1568 | 0.0159 | 0.0250 | 0.0382 | 0.1456 | 0.1456 | 0.1713 | 0.2602 | 0.3989 |
| | CoLLM | 0.1298 | 0.1344 | 0.1388 | 0.1602 | 0.0173 | 0.0213 | 0.0296 | 0.0444 | 0.1658 | 0.1880 | 0.2895 | 0.4058 |
| | HLLM | 0.1285 | 0.1351 | 0.1457 | 0.1668 | 0.0189 | 0.0232 | 0.0352 | 0.0528 | 0.1652 | 0.1933 | 0.2920 | 0.4037 |
| Ours | Molar | 0.1407 | 01478 | 0.1580 | 0.1773 | 0.0197 | 0.0242 | 0.0359 | 0.0539 | 0.1768 | 0.2068 | 0.3124 | 0.4320 |

## 4.2 RESULTS

In this section, we compare Molar against traditional, content-based, and llm-based baselines, taking into metrics of both NDCG and Recall on PixelRec, MovieLens, and Amazon datasets, to showcase the effectiveness and robustness of Molar.

**Main Results.** We implemented the Molar framework on three datasets. The comparison with baseline models is summarized in Table 1. Key observations are as follows:

**(a) Superior Performance Across Datasets.** Molar consistently outperforms all baseline models across three datasets. Our method achieves over a 7% improvement in NDCG and Recall metrics on the MovieLens dataset, with similar enhancements on PixelRec and Amazon datasets compared to the strongest baseline. This demonstrates Molar effectively integrates traditional sequential information with the expansive knowledge and reasoning capabilities of LLMs. By leveraging user interaction sequences, Molar retains the strengths of collaborative filtering to capture user behavior patterns, while using MLLMs' advanced visual and language understanding to interpret complex user intents and contextual nuances. The synergy between sequential data and LLM-driven insights allows Molar to balance explicit user preferences with implicit, semantically rich information, enhancing recommendation relevance and accuracy.

**(b) Enhanced Multimodal Integration.** By efficiently integrating textual and visual features through MLLM, Molar achieves a substantial improvement in recommendation quality. Unlike SASRec-Content, which separately uses Vision Transformers (Vit) (Dosovitskiy, 2020) for images and Bert (Devlin, 2018) for text, Molar leverages the MLLM as MIRM. This integrated approach results in a remarkable 44% improvement due to better item embedding extraction compared to SASRec-Content. The primary reason for this boost is the pretraining capabilities of MLLMs, which align and fuse visual and textual representations. This alignment allows for a deeper understanding of item features, streamlining the information processing pipeline and significantly enhancing the quality of the embeddings, thus improving recommendation performance.

**(c) Combining Semantics with Collaboration.** While methods like HLLM and CoLLM improve on ID-based and content-based approaches, they fall short compared to Molar, particularly in NDCG metrics, due to their inability to capture collaborative knowledge essential for recommendations. LLMs excel in processing textual data but struggle with user-user and item-item interactions found in traditional systems. Molar bridges this gap by combining LLMs' semantic understanding with the collaborative filtering strengths of traditional methods, enabling it to use both deep semantic insights and relational knowledge for more accurate and relevant recommendations.

**Impact of DUEG.** We conducted experiments to evaluate various representation methods for DUEGs, including FPMC, SASRec, GRU4Rec, and our proposed LLM Qwen2vl backbone as the DUEG. As shown in Figure 3a, the results indicate that Molar, using an LLM backbone as the DUEG,

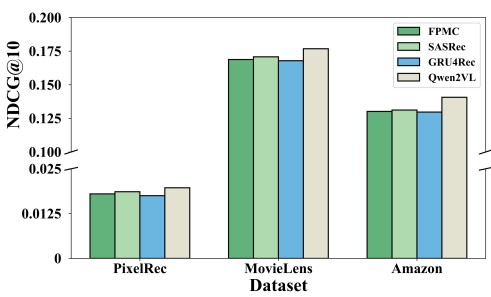 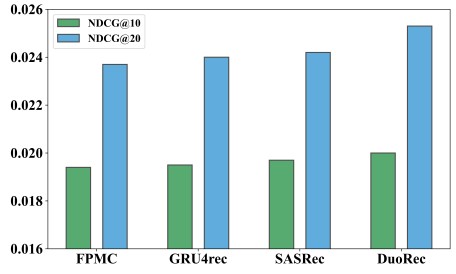

(a) **Performance comparison of different DUEGs.** Qwen2vl-2b is used as MIRM for all. (b) **Comparison of different post-alignment models for contrastive learning on PixelRec.**

Figure 3: Performance comparisons of different DUEGs and post-alignment models

outperforms all other methods across the three datasets. This not only validates the effectiveness of our innovative user representation approach but also highlights the limitations of relying solely on textual and visual information (e.g., text and image metadata) or sequential information (e.g., behavioral ID tokens). Compared to the highest-performing baseline, SASRec, Molar demonstrates an average improvement of 6.0% on the PixelRec dataset and an even more substantial average improvement of 7.2% on the Amazon dataset. The superior performance of Molar can be attributed to the extensive pretraining of the LLM backbone, which imbues it with comprehensive world knowledge. Additionally, the alignment between fine-tuning and recommendation systems allows the training process to converge with only a few epochs (5 epochs), whereas other non-LLM baselines require prolonged training periods (e.g., 200 epochs for SASRec) to achieve convergence.

**Impact of Post-Alignment Models.** In the process of post-alignment contrastive learning, integrating ID information results in varying degrees of improvement across different traditional sequential recommendation models. To verify the performance impact brought by different traditional sequential recommendation models, we conducted the experiments shown in Figure 3b. The results reveal a clear trend: stronger sequential recommendation models like DuoRec better support post-alignment contrastive learning, enhancing the integration of ID-based collaborative filtering signals into LLMs. This integration significantly boosts recommendation accuracy, coverage, and performance. DuoRec's robust architecture captures richer user-item interaction patterns, enabling the LLM to leverage nuanced ID information for top NDCG and Recall scores, which highlight the importance of selecting powerful sequential models for contrastive learning, as they refine the process and ensure coherent ID integration, unlocking the full potential of collaborative filtering for diverse scenarios.

**Impact of Input Data Modality.** To gain a thorough understanding of how various data modalities influence the performance of Molar, particularly the contribution of multimodal fusion and integration of knowledge from various modalities, we conducted an in-depth analysis on PixelRec in Table 2.

Our findings demonstrate that multimodal fusion enhances recommendation performance. The fusion of multiple modalities significantly enhances recommendation performance. A comparative analysis of Molar on three input types reveals that the combined text with image input yields the best results. This can be attributed to the unique knowledge contributed by each modality, which cannot be captured by the others. MLLM effectively integrates this complementary information, demonstrating its potential as a robust foundation for multimodal recommendation tasks.

|  | N@10 | N@20 | N@50 | R@10 | R@20 | R@50 |
|---|---|---|---|---|---|---|
| Image Only | 0.0182 | 0.0217 | 0.0292 | 0.0329 | 0.0512 | 0.0858 |
| Text Only | 0.0181 | 0.0228 | 0.0296 | 0.0335 | 0.0514 | 0.0860 |
| Image + Text | **0.0197** | **0.0242** | **0.0313** | **0.0359** | **0.0539** | **0.0895** |

Table 2: **Benefit of different modality inputs.**

**Ablation Study.** To analyze the contributions of components in the proposed Molar method, particularly the fine-tuning of the MIRM on multimodal data and the post-alignment for user representation, we conducted an ablation study on the PixelRec dataset.

**(a) Effect of Fine-Tuning Data for MIRM.** The fine-tuning data for MIRM comprises three key components: **Image-Text (IT)** where the input is item images and the output is detailed textual

descriptions; **Structured Attributes (SA)** where the input is item title and metadata and the output is detailed descriptions; **User Behavior (UB)** where the input is historical item interactions (descriptions and images) and the output is predicted subsequent items. To evaluate the contribution of each fine-tuning data component, we systematically removed one type of fine-tuning data at a time during stage 1. Additionally, removing all three types (*w/o ALL*) effectively disables the fine-tuning stage. The results, presented in Table 3, show that the model achieves its best performance when all three fine-tuning data types are used together. Conversely, removing any single type of data leads to a noticeable decline in performance, highlighting the importance of each component in enhancing the model's overall effectiveness. Interestingly, the performance when structured attributes (*w/o SA*) are excluded remains higher than when either image-text (*w/o IT*) or user behavior (*w/o UB*) data are omitted. Even in the absence of structured attributes, the combination of image-text and user behavior data can effectively fine-tune MIRM, enabling it to learn and align meaningful representations.

**(b) Impact of Post-Alignment Contrastive Learning.** To examine the role of post-alignment in user representation, we conducted an ablation experiment by removing the post-alignment contrastive learning (*w/o CL*) module. As shown in Table 3, removing the contrastive learning module significantly decreases performance, particularly in the NDCG metric. This highlights the critical importance of incorporating ID-based information and collaborative signals. Post-alignment ensures effective feature-level communication and inter-

|  | N@10 | N@20 | N@50 | R@10 | R@20 | R@50 |
|---|---|---|---|---|---|---|
| *Full Model* | | | | | | |
| **Molar** | **0.0197** | **0.0242** | **0.0313** | **0.0359** | **0.0539** | **0.0895** |
| *Fine-Tuning Data* | | | | | | |
| *w/o IT* | 0.0186 | 0.0227 | 0.0298 | 0.0339 | 0.0512 | 0.0841 |
| *w/o SA* | 0.0189 | 0.0237 | 0.0302 | 0.0349 | 0.0528 | 0.0859 |
| *w/o UB* | 0.0183 | 0.0220 | 0.0287 | 0.0324 | 0.0495 | 0.0828 |
| *w/o ALL* | 0.0180 | 0.0219 | 0.0285 | 0.0313 | 0.0479 | 0.0808 |
| *Post-Alignment* | | | | | | |
| *w/o CL* | 0.0182 | 0.0225 | 0.0294 | 0.0325 | 0.0496 | 0.0819 |

Table 3: **Ablation study on the PixelRec dataset.**

action between ID-based models and content-based MLLM, enabling the model to fully leverage multimodal information in a collaborative filtering context.

**Impact of Different MLLM Backbone.** To evaluate the impact of MLLM backbones on Molar, we compare models with varying architectures and sizes, using LoRA for 7B+ models due to computational constraints. As shown in Table 4, Qwen2vl outperforms others at the same scale, indicating that backbone choice significantly influences performance. Performance improves with model size, and the 7B Qwen2vl surpasses its 2B variant even under LoRA, demonstrating the benefits of both scale and architectural strengths. These results highlight that while larger models help, backbone selection is equally critical—especially under resource constraints.

| MLLM Backbone | Training Type | N@10 | N@20 | R@10 | R@20 |
|---|---|---|---|---|---|
| Qwen2-VL-2B | Full-tuning | 0.0197 | 0.0242 | 0.0359 | 0.0539 |
| InternVL2.5-2B [2] | Full-tuning | 0.0191 | 0.0237 | 0.0349 | 0.0521 |
| deepseek-vl-1.3b [3] | Full-tuning | 0.0183 | 0.0225 | 0.0334 | 0.0499 |
| Qwen2-VL-7B | LoRA | **0.0200** | **0.0251** | **0.0369** | **0.0555** |
| Llama-3.2-11B-Vision [4] | LoRA | 0.0194 | 0.0249 | 0.0357 | 0.0542 |

Table 4: Comparison of Different MLLM Backbone.

## 5 CONCLUSION

In this paper, we introduced **Molar**, a novel framework for sequential recommendation that bridges the gap between collaborative filtering and multimodal content modeling using LLMs. While traditional LLM-based approaches excel in semantic understanding, they lack the ability to incorporate collaborative filtering signals, limiting their recommendation performance. To overcome this limitation, **Molar** integrates multimodal data with ID-based collaborative signals, leveraging an MLLM to generate unified item representations and a post-alignment mechanism to align user embeddings effectively. By combining the strengths of multimodal content modeling and collaborative filtering, **Molar** captures both user interests and contextual semantics, enabling precise and personalized recommendations. Extensive experimental results demonstrate that **Molar** consistently outperforms traditional methods and state-of-the-art LLM-based baselines, validating its ability to fully exploit multimodal data and collaborative signals for sequential recommendation tasks.

**Ethics statement.** We confirm that this work aligns with accepted ethical standards in machine learning research. All data and methodologies used are publicly available or properly cited.

**Reproducibility statement.** To support reproducibility, we have provided full details of our experimental setup, including hyperparameters and dataset descriptions, in the experimental paragraph. Code is available.

**The use of large language models (LLMs).** We utilize LLMs to assist and enhance our writing. They help us improve the quality and effectiveness of our textual expression.

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

**Appendix**

**Statistics of Datasets.**   We presents the statistical characteristics of our datasets, shown in table 5.

| Dataset | Amazon | PixelRec | MovieLens |
|---|---|---|---|
| # User | 993,087 | 50,000 | 6,040 |
| # Item | 301,312 | 82,864 | 3,706 |
| # Interaction | 8,813,442 | 989,476 | 1,000,209 |

Table 5: Statistics of Datasets.

**Broader impact.**   This paper presents work whose goal is to advance the field of Machine Learning. There are many potential societal consequences of our work, none which we feel must be specifically highlighted here.

**Limitation.**   While Molar effectively integrates multimodal large language models (MLLMs) into sequential recommendation tasks, several limitations remain. First, the framework requires multi-task fine-tuning to optimize multimodal representations, which can be time-intensive and may hinder its deployment in real-time applications. Second, due to computational constraints, we are unable to train larger language models, and the quality of the generated multimodal item representations heavily depends on the underlying capabilities of the MLLMs. If the base models are suboptimal, the overall recommendation performance may degrade. In future work, we aim to develop an end-to-end training framework and incorporate more advanced MLLMs with larger parameter sizes to enhance the quality of generated representations, thereby improving the overall efficacy of Molar.

