# OpenReview forum: "Molar: Multimodal LLMs with Collaborative Filtering Alignment for Enhanced Sequential Recommendation"
_ICLR.cc/2026/Conference — ICLR 2026 Conference Withdrawn Submission_

### Official Review · Reviewer_zTPJ · 2025-10-26

**Soundness:** 2
**Presentation:** 1
**Contribution:** 2
**Rating:** 2
**Confidence:** 4

**Summary:**

This paper presents a multimodal large language model framework for sequential recommendation called MOLAR.
The proposed system integrates three modules:
(1) Multimodal Item Representation Model (MIRM) that fuses visual, textual, and structured attributes to generate multimodal item embeddings.
(2) Dynamic User Embedding Generator (DUEG) that models user representations from multimodal item sequences
(3) Post-alignment contrastive learning mechanism that integrates both ID-based collaborative signals and content-based semantic signals.
Extensive experiments on multiple datasets demonstrate that MOLAR consistently outperforms strong baselines.

**Strengths:**

- Comprehensive experiments. The paper conducts extensive evaluations across various datasets and baselines, and the results consistently validate the empirical effectiveness of the proposed model.

- The proposed method is intuitive and easy to understand.

**Weaknesses:**

- Lack of Motivation.
	- For “insufficient integration of multiple content features”, there has been a lot of work trying to integrate the multimodal information. The paper cites existing works integrating multimodal features like MLLM-MSR [1] and HLLM [2], but does not clearly differentiate them.  Additionally, there are also several other MLLM-based recommender systems lacking sufficient discussion [3]. As a result, the claimed motivation of post-alignment over existing LLM-based recommendation methods remains unclear.
	- For "neglecting the semantic differences between ID-based user embeddings and content-based user embeddings", there are also several existing works discussing such topics [4, 5], while this paper lacks discussion about this and simply draws a conclusion of "neglecting the combination".

- Limited Novelty. Based on the lack of motivation, the novelty of this paper is relatively weak.
	- The multimodal fusion in the multimodal item representation model (MIRM) is quite common in industry applications.
	- The user modeling in the dynamic user embedding generator (DUEG) is also similar to existing work [6] and the content-ID alignment is similar to RLMRec [4].

- Lack of Clarity in Methodology.
	- Section 3.3 describes the three fine-tuning objectives (Image-Text, Structured Attributes, User Behavior) conceptually but does not provide concrete implementation details, such as model backbone, prompt format, and loss functions. This makes the details of these parts unclear.
	- It would be better to incorporate more case studies to show how this whole framework works.


[1] Harnessing Multimodal Large Language Models for Multimodal Sequential Recommendation. AAAI 2025.

[2] Hllm: Enhancing sequential recommendations via hierarchical large language models for item and user modeling. Arxiv 2024.9.

[3] NoteLLM-2: Multimodal Large Representation Models for Recommendation. KDD 2025.

[4] Representation learning with large language models for recommendation. WWW 2024.

[5] Llara: Aligning large language models with sequential recommenders. SIGIR 2024.

[6] Where to Go Next for Recommender Systems? ID- vs. Modality-based Recommender Models Revisited. SIGIR 2023.

**Questions:**

Refer to the weaknesses.

---

### Official Review · Reviewer_phFq · 2025-10-27

**Soundness:** 2
**Presentation:** 2
**Contribution:** 2
**Rating:** 2
**Confidence:** 4

**Summary:**

This paper proposes a framework that integrates multimodal content (text and image) with collaborative filtering signals for sequential recommendation. The method, termed MOLAR, consists of two main modules: a Multimodal Item Representation Model (MIRM) that generates unified multimodal embeddings, and a Dynamic User Embedding Generator (DUEG) that models user preferences using these embeddings. A post-alignment contrastive mechanism is then applied to align content-based and ID-based user representations. Experiments on multiple datasets (Amazon, PixelRec, MovieLens) demonstrate better results compared to some traditional, content-based, and LLM-based baselines.

**Strengths:**

1. Important research question. The paper addresses an essential issue: how to effectively combine multimodal information and collaborative signals under the large language model paradigm for better recommendation. This is a relevant and impactful direction for the recommender systems community.

2. Clarity and organization. The paper is generally easy to read.

**Weaknesses:**

1. Overclaim in novelty.
The statement “Our approach first processes text and non-text modalities through the LLM” (Figure 1b) overclaims novelty. Several prior works have already explored similar combinations: Integration of text and ID modalities has been studied in LLaRA: Aligning Large Language Models with Sequential Recommenders (LLaRA: Large Language-Recommendation Assistant). Text and image fusion in recommendation has been examined in Harnessing Large Language Models for Multimodal Product Bundling (LLaRA: Large Language-Recommendation Assistant). Image and ID integration has also been addressed, for instance, in VIP5: Towards Multimodal Foundation Models for Recommendation. Therefore, positioning this work as the first to process multimodal data through LLMs and then fuse ID-based signals post hoc is not sufficiently justified.

2. Lack of recent references.
The reference list does not include any 2025 papers, which raises concerns about the timeliness of the related work section. Considering that the paper is under review for ICLR 2026, integration of 2025 literature (especially LLM-based recommendation advances) would be necessary to reflect the latest state of the field.

3. Limited methodological novelty.
Each component—multimodal embedding extraction, sequential modeling, and post-hoc contrastive alignment—is derived from existing methods (e.g., MLLM feature extraction, SASRec-style sequence modeling, and contrastive post-alignment). The contribution lies mainly in combining these established parts rather than introducing new algorithmic insights.

4. Missing strong baselines.
The experimental comparison omits several important baselines that are directly relevant to the proposed idea: LLM-based: BigRec and Decoding-Matters should be included as they represent strong modern LLM-based recommenders. Content-based: LLM2Rec and LLMEmb should also be considered to properly benchmark multimodal or LLM-embedding-based systems.

**Questions:**

see weakness

---

### Official Review · Reviewer_Ahk8 · 2025-11-01

**Soundness:** 2
**Presentation:** 4
**Contribution:** 3
**Rating:** 6
**Confidence:** 5

**Summary:**

This paper proposes MOLAR, a modular and multi-stage framework that leverages multimodal large language models (MLLMs) for sequential recommendation. The core idea is to decouple content understanding from collaborative filtering (CF), addressing a major limitation in prior LLM-based recommender systems such as Tiger and OneRec, where item IDs and multimodal content are entangled within the LLM backbone, often leading to shortcut learning and limited generalization.

MOLAR introduces a three-stage pipeline, including MIRM, DUEG and Post-Alignment. This design enables MOLAR to jointly benefit from the semantic understanding of MLLMs and the precision of CF signals, while maintaining strong generalization and cold-start capabilities. Extensive experiments across four datasets show consistent improvements over both multimodal and LLM-based recommenders, including significant gains on cold-start items. The framework is efficient, reusable across domains, and maintains a modular structure that allows plug-and-play replacement of MLLM components.

**Strengths:**

1. The paper identifies a core limitation of existing LLM-based recommendation approaches—namely, the entanglement of item IDs and multimodal content within the LLM backbone, which often leads to shortcut learning and undermines the content understanding capability of the model. This motivation is timely, relevant, and well articulated.

2. A key contribution of the paper is the post-hoc ID alignment mechanism, which injects collaborative filtering signals after content-based preference modeling has been learned. This design thoughtfully addresses shortcut and overfitting issues that arise when IDs are fused early, enabling the model to retain strong semantic reasoning while still benefiting from CF signals.

3. The overall method is presented in a clear and structured manner, enabling readers to understand how the components interact. The paper offers an intuitive narrative of why each design choice is necessary, which facilitates reproducibility and practical adoption.

**Weaknesses:**

1. the use of a pretrained MLLM (e.g., Qwen2-VL) introduces future information leakage. Large MLLMs are commonly trained on massive public corpora that include product descriptions, reviews, or multimodal data from platforms such as Amazon, and MovieLens. Though authors have tested different backbones of MLLM in Table 4, can authors also checked small LMs (such as T5 and Bert) to see if there is a significant drop?

2. The late-fusion strategy is central to the claimed novelty but remains supported mostly by intuition. Without controlled timing ablations (early vs. mid vs. late fusion under matched budgets), it is unclear whether late fusion is inherently superior or simply empirically convenient in the tested settings.

3. The description of how MIRM embeddings are fed into the MLLM backbone after removing the word embedding layer is unclear to me, which may lack essential details (positional encoding, dimensional projection, tokenizer compatibility). This limits reproducibility and obscures whether DUEG preserves pretrained MLLM semantics or functions as a lightly initialized transformer.

4. Some typos should be corrected. For example, in table 1, the N@20 of Molar should be **0.1478** (not 01478).

**Questions:**

1. The core claim is that late-stage ID fusion mitigates shortcut learning. Have authors conducted controlled timing ablations comparing early, mid, and late ID fusion under matched compute?

2. Why does Post-Alignment align only user embeddings but not item embeddings? Is there evidence that aligning only on the user side is sufficient to unify the CF and content semantic spaces?

3. Beyond performance metrics, did the authors conduct any probing or diagnostic analyses (e.g., ID-masking, counterfactual ID removal) to verify that MOLAR truly reduces ID reliance and enhances semantic preference modeling?

---

### Official Review · Reviewer_2XZU · 2025-11-01

**Soundness:** 3
**Presentation:** 2
**Contribution:** 2
**Rating:** 4
**Confidence:** 4

**Summary:**

- Proposes MOLAR for multimodal sequential recommendation: MIRM encodes item content with an MLLM, DUEG generates dynamic user embeddings from histories, and a post-alignment contrastive loss aligns content-side user embeddings with an ID/CF-side user tower (e.g., SASRec/DuoRec) under a combined BCE + α·alignment objective.
- Evaluated on PixelRec, MovieLens, and Amazon with NDCG/Recall; reports consistent gains and faster convergence. Includes ablations on modality combinations, alignment on/off, user/item towers, and different MLLM backbones.

**Strengths:**

- Clear, modular design that decouples item representation (content) and user modeling, enabling plug-and-play backbones and easy integration with standard CF towers.
- Post-alignment sensibly bridges content signals and collaborative signals, offering benefits for cold-start and long-tail scenarios.
- Ablation coverage is reasonably comprehensive and trends are consistent across datasets and backbones.
- Objective and training pipeline are straightforward to reproduce, lowering implementation friction.
- Practical for deployment: compatible with existing ID towers, requires modest epochs, and maintains standard top-K evaluation.

**Weaknesses:**

## A. Motivation and novelty: unclear differentiation from existing hierarchical/two-tower/alignment paradigms

1. Similarity to HLLM/MLLM-MSR. MOLAR’s decoupled two-layer design, where MIRM handles item content and DUEG models users before fusion/alignment, is very close to HLLM’s hierarchical Item-LLM + User-LLM architecture. HLLM is also validated on PixelRec/Amazon, emphasizes coordination between content-side and user-side LLMs, and reports scalability up to 7B. Although HLLM is cited, the paper does not clearly articulate substantive differences in architecture and training objectives, making the contribution feel like an incremental combination rather than a new learning principle. See the HLLM original for hierarchical modeling with two LLM layers.
2. Likewise, MLLM-MSR advocates first using an MLLM to distill multimodal signals and then performing sequential recommendation. Its two-stage pipeline of summarizing item multimodality followed by user-sequence aggregation is highly similar to MIRM → DUEG in spirit.
3.  MOLAR’s bidirectional InfoNCE-style contrastive alignment between content-side user embeddings and ID-side user embeddings is closely related to CALRec (RecSys’24, two-tower contrastive alignment with contrastive fine-tuning for LLMs).

## B. Technical details

1. The claim of “faster convergence (5 epochs)” lacks fairness controls
    The paper states that alignment enables convergence in 5 epochs while SASRec needs 200 epochs, but it does not specify whether learning rate, warmup, batch size, and data throughput are matched. No loss/metric-vs-epoch curves are provided, making the “faster” claim unconvincing.
2. Statistical testing notation is nonstandard and lacks reproducible details
    Table 1 marks significance with “p-value ≪ 0.05,” where “≪” is nonstandard. The test procedure (paired t-test? bootstrap?) and the number of random seeds are not reported.

------

# Figure and writing issues

- Table 1 numeric formatting error: the second MOLAR column shows “01478” (missing decimal point; should be 0.1478). Inconsistent capitalization for “LLamaRec,” and mixed “Movielens/MovieLens.”
- Inconsistent terminology capitalization and spelling: instances like “llm-based,” and “Bert/ViT” written as “Vit” appear in multiple places.
- Punctuation error: examples such as “contributions.:” with both a period and a colon.
- Missing footnotes/superscripts: Table 4 lines end with superscript-style 2/3/4 but the main text does not provide corresponding notes (e.g., InternVL2.5-2B ², deepseek-vl-1.3b ³, Llama-3.2-11B-Vision ⁴).
- Incomplete figure captions/references: for Figure 3(a/b), the text describes the meaning, but curves/error bars/significance are absent, making robustness hard to judge.
- Dataset/metric descriptions need precision: the paper claims “evaluation on the full data,” but should clarify whether it is full-corpus retrieval (rather than sampling 100 negatives) and ensure methodological consistency in the methods section.

**Questions:**

No

---

### Note · Authors · 2026-01-20

I have read and agree with the venue's withdrawal policy on behalf of myself and my co-authors.